# Evaluating the impact of injury prevention interventions in child and adolescent sports using the RE-AIM framework and CERT: A systematic review

Thomas Hughes[1]*, John O'Hara[1], Alan Burton[1], Nick London[1,2], Stacey Emmonds[1]

**1** Carnegie School of Sport, Leeds Beckett University, Leeds, United Kingdom, **2** The Yorkshire Knee Clinic, Nuffield Health Leeds Hospital, Leeds, United Kingdom

⊙ These authors contributed equally to this work.
* Thomas.hughes@leedsbeckett.ac.uk

**Data Availability Statement:** All relevant data are within the paper and its Supporting Information files

## Abstract

### Background

Participation in sport is a popular pastime for children and adolescents that improves their physical health, mental health and motor skills. Musculoskeletal injuries are a relatively common downside of sports participation and can have negative long-term consequences. Injury prevention programmes have demonstrated effectiveness in child and adolescent sports, provided compliance is adequate. However, little is known about the factors which relate to their impact on the wider community and whether the prevention programmes have been adopted and maintained in the long-term. The objective of this review was to assess the current literature on exercise-based injury prevention interventions in child and adolescent sports (aged under 19 years) against the *'Reach'*, *'Effectiveness'*, *'Adoption'*, *'Implementation'*, *'Maintenance'* (RE-AIM) framework and Consensus of Exercise Reporting Template (CERT), to ascertain level of reporting for the components which relate to external validity.

### Methods

Seven electronic databases; PubMed, Medline, SPORTDiscus, PsycINFO, CINAHL, Scopus and The Cochrane Library, were searched from date of inception to July 2022 using the themes of: '*Child and Adolescent*', '*Sport*', '*Injury*' and '*Prevention*'. Eligibility criteria included: Experimental trial design, exercise-based intervention programmes, primary outcome of injury incidence and participants aged under 19 years. Two reviewers assessed each trial independently against the RE-AIM model dimension items checklist (RE-AIM MDIC) and Consensus on Exercise Reporting Template (CERT) before reaching a consensus on reporting.

**Funding:** Funding for this PhD project was jointly provided by the Carnegie School of Sport, Leeds Beckett University, Leeds, United Kingdom https://www.leedsbeckett.ac.uk/carnegie-school-of-sport/ Zimmer Biomet, Warsagie Indiana, United States https://www.zimmerbiomet.com/en and the Yorkshire Knee Clinic, Leeds, United Kingdom. https://yorkshirekneeclinic.com/ The funders had no role in the study design, data collection and analysis, decision to publish, or preparation of the manuscript.

**Competing interests:** The authors have declared that no competing interests exist.

## Results

Forty-five unique trials met the eligibility criteria. Mean reporting level for all studies across the whole RE-AIM MDIC was 31% (SD ± 16.2%, Range 7–77%). The domain of *'effectiveness'* was the most comprehensively reported (60%), followed by *'implementation'* (48%), *'reach'* (38%), *'adoption'* (26%) and *'maintenance'* (7%). The mean reporting score for the CERT was 50% (SD ± 20.8, range 0–81%).

## Conclusion

Published data on injury prevention in child and adolescent sports is highly focussed on the effectiveness of the intervention, with little consideration given to how it will be adopted and maintained in the long-term. This has led to considerable gaps in knowledge regarding optimal programme implementation, with a lack of data on adoption and maintenance contributing to the gap between research and practice. Future research needs to place greater focus on external validity and consider incorporating the study of implementation and feasibility as part of effectiveness trial design. This approach should provide the data that will help narrow the considerable gap between science and practice.

## Trial registration

PROSPERO Registration number CRD42021272847.

## Introduction

Sport participation is a popular pastime for children and adolescents, with 68% of children aged 7–11 and 63% of adolescents aged 11–16 reporting participation in organised team sports within the United Kingdom [1]. Sport participation plays an important role in the health and development of children, with those regularly engaging in sports and physical activity at a lower risk of obesity [2], displaying improved markers of cardiovascular fitness [3] and demonstrating greater development of motor skills [4]. The benefits are not just physical, sports participation also has the potential to improve mental health in adolescents [5].

Sports-related injuries in children and adolescents who engage in organised sport are a relatively common occurrence, with an incidence that is typically in the range of 1–10 injuries per 1000 hours of participation, depending on sport and sex [6]. Sports injury in children and adolescents can have long-term negative health consequences; physeal injuries in adolescents can disrupt bone growth [7] and anterior cruciate ligament (ACL) rupture can lead to early onset osteoarthritis [8], therefore sports injury prevention should be viewed as a priority.

There are a growing number of different sports injury prevention programmes, which come under the definition of 'neuromuscular training', with pooled analysis indicating that these programmes are effective in reducing sports injury in children and adolescents taking part in organised sport by approximately 40–50% [9, 10]. These programmes have been tailored to specific sports such as the *'FIFA 11/FIFA 11+'* for soccer (FIFA/F-MARC, [11]), *'Activate'* for schoolboy rugby [12], *'SHRed'* for basketball [13] and *'VolleyVeilig'* for volleyball [14]. The *'FIFA 11+'* has also been adapted for younger children, aged seven to 13, with the resulting programme *'11+ Kids'* demonstrating effectiveness for reducing injury rates in this age group by 48% [15]. Other programmes have targeted anatomical regions, such as *'HarmoKnee'* [16]

or specific injuries, such as the '*Prevent Injury and Enhance Performance Program' (PEP)* targeted at reducing risk of ACL injury (Santa Monica Sports Medicine Research Foundation [17]). Whilst many of these programmes have demonstrated effectiveness, this has depended on adequate compliance [18–20] and often a high frequency of delivery [12].

Understanding how to promote compliance and high frequency of delivery falls within the field of implementation science, which has been defined as 'the scientific study of methods to promote the systematic uptake of research findings and other evidence-based practices into routine practice, and, hence, to improve the quality and effectiveness of health services' [21]. The use of implementation science in injury prevention programme design appears lacking, according to previous reviews in this area [22, 23] and has led to a considerable gap between research and practice [24].

The '*Reach', 'Effectiveness', 'Adoption', 'Implementation', 'Maintenance'* (RE-AIM) framework [25], was initially proposed in 1999 by Glasgow and colleagues with the aim of improving the translation of public health research into practice. The RE-AIM framework has been applied throughout the intervention trial process from grant proposal appraisal [26] to informing trial design and study evaluation [27]. In the field of sports injury prevention, the framework was used to inform the design and subsequent implementation of a study on the effectiveness of the '*Activate*' programme for rugby injury prevention in UK schools [28]. Utilising the RE-AIM Model Dimension Items Checklist (RE-AIM MDIC) [26] the framework has been used as a tool for the systematic evaluation of public health intervention trials [29, 30] and adapted for use in a sports injury setting to evaluate the reporting of trials aimed at injury prevention [22, 23]. Each domain within the RE-AIM framework has a specific definition which must be considered when applying the framework to a particular context. The domain of '*reach*' covers how the study participants relate the target population; the '*effectiveness*' domain refers to how the intervention impacts on health outcomes; '*adoption*' concerns the settings and individuals who deliver the intervention; '*implementation*' addresses how the intervention was delivered and '*maintenance*' concerns the sustainability of the intervention over the long-term [25]. Although the RE-AIM framework considers elements of intervention design as part of the '*implementation*' domain it was not developed specifically for exercise-based interventions and therefore lacks the level of detail required to appraise the reporting of an exercise programme, a weakness noted in the use of the RE-AIM framework in this context [23]. A recently developed framework for assessment of exercise interventions the Consensus of Exercise Reporting Template (CERT) [31] provides detailed assessment on reporting of the exercise programme details within an intervention trial. It has been utilised in multiple systematic reviews as a method of reporting assessment [32], demonstrating good levels of inter-rater reliability [33].

Previous reviews employing the RE-AIM framework to assess sports injury prevention studies [22, 23] identified significant gaps in '*adoption*' and '*maintenance*'. Their systematic evaluations of the literature indicated that research was heavily focussed on '*effectiveness*', and '*implementation*', often leaving the other domains neglected. Publishing robust data on effectiveness, whilst neglecting long-term measures, demonstrates the conflict between promoting internal or external validity in study design [34]. Study designs that aim for high internal validity, i.e., the extent to which the results are true for the studied population [35], often compromise on external validity, the extent to which the results in the study reflect what can be expected outside of the study population [36]. This conflict has historically been managed in the research cycle by conducting efficacy studies with high internal validity to establish causality, followed by implementation and feasibility studies to establish external validity [37]. This sequential process is time consuming and often results in a considerable time lag between demonstrating effectiveness and translation into clinical practice [38]. This lag time is a potent

contributor to the gap between science and practice, supporting the recommendation to include elements of implementation and feasibility in the initial trial design [39]. One potential approach is to study effectiveness and implementation concurrently as part of a hybrid design [40]. By measuring the effectiveness of relevant variables at the same time as the implementation strategy it may be possible to understand what dictates successful implementation and how that impacts on injury outcomes [38].

Previous systematic reviews have assessed the effectiveness of neuromuscular based injury prevention programmes in child and adolescent sports [9] and used meta-analytic approaches to understand the influence of sex and exercise types on effectiveness [10]. Reviews have also studied the influence of compliance [41] and most recently the effectiveness of warm-up specific interventions [42]. These reviews give us ample data on the effectiveness of intervention programmes and the internal validity of the studies but tell us little regarding the external validity. There are currently no reviews that have systematically assessed injury prevention trials in child and adolescent sports specifically for their reporting of factors which relate to external validity and the potential impact on the wider population. Therefore, the objective of this review is to evaluate the published literature on sports injury prevention interventions in child and adolescent organised sports (under 19 years of age), using the RE-AIM framework [25] and CERT [31]. Through this method we aim to identify areas that are frequently neglected in the reporting of intervention studies that undermine their external validity and potential impact on the wider population. We also aim to highlight positive practices that if universally adopted would help promote the translation of science to practice.

## Methods

### Search strategy

This systematic review was registered with PROSPERO (Registration number CRD42021272847) and conducted in accordance with the Preferred Reporting Items for Systematic Reviews and Meta-Analysis (PRISMA) guidelines and 2020 PRISMA statement [43].

It was conducted with ethical approval from [**insert institution post review**]. Seven electronic databases (PubMed, Medline, SPORTDiscus, PsycINFO, CINAHL, Scopus, Cochrane Library) were searched from date of inception to July 2022 by one reviewer (TH). Additional articles were obtained through searching of reference lists of similar publications, contact with experts in the field and targeted searching of known authors for follow-up papers providing further reporting on previous trials. An updated search was conducted in May 2023 to identify additional articles. Four key search themes were identified: '*Child and Adolescent*', '*Sport*', '*Injury*' and '*Prevention*'. These four themes were expanded to include all potential variations and searches were conducted with text words and medical subject headings (MeSH). The full search strategy has been published online via PROSPERO and can be found in the S1 Table. Following searches in each database the references were exported and collated in Endnote©, duplicates were then removed, and the references were exported to Microsoft Excel© for screening.

Following retrieval, sorting and duplicate removal the titles and abstracts were independently screened by two reviewers (TH and AB) with potentially relevant articles brought forward for full text screening against the inclusion criteria (detailed below). Any disagreements in article inclusion between reviewers were discussed and if a consensus was not reached, the relevant articles were brought forward for full text screening. Any disputes occurring following full text screening were referred to a third Author (JO) for resolution.

For certain elements of the RE-AIM framework such as long-term maintenance it was possible that subsequent articles reporting this data were published following the publication of

the original trial. Further targeted searching was required to identify these articles and assess them in conjunction with the original trial. During screening any articles identified that related to a previous trial were brought through to full text screening. Once original trials had been identified each named author's subsequent publication history was searched specifically on PubMed, Google Scholar and Scopus to identify any further articles pertaining to the original trial.

## Eligibility criteria

Articles were included if they satisfied all of the following inclusion criteria; (1) Original data with full text available (Full text article published); (2) English language; (3) Peer reviewed; (4) Quantitative outcome of musculoskeletal sports injury; (5) Investigated the impact of an exercise-based injury intervention; (6) Prospective analytical design including control/comparator group (e.g., Randomised controlled trial, quasi-experimental/cohort study); (7) Included sports participants aged 18 years or younger. Studies were excluded if they were; (1) Incomplete studies or manuscripts; (2) Full text not in the English language or unavailable; (3) Not musculoskeletal injury e.g., concussion; (unless concussion data was analysed separately) (4) Participants with underlying health conditions or existing injuries at the time of study; (5) Inclusion of any participants aged 19 years or older; (6) Interventions utilising worn hardware e.g., orthotics. (7) study of risk factors for injury without injury incidence data.

For studies with participant ages reported as mean age, without range, indicating the possible inclusion of participants aged 19 years and older, the study authors were contacted for the data on age range of participants. This occurred on two occasions with both authors responding, this resulted in one paper being removed due to inclusion of 19-year-olds [44] and one paper included as all participants were aged under 19 years [45].

## Data extraction and risk of bias assessment

Article data was extracted from each article by one reviewer (TH) including: authors, year of study, study design, demographics of participants (including age and gender), sample size, setting of the intervention, country, sport, intervention type, and session details. Risk of bias assessment was carried out in accordance with the Cochrane Review guidelines utilising the recently revised RoB 2 tool [46] for randomised controlled trials. The ROBINS-I tool was used for non-randomised trials [47].

## Assessment with the RE-AIM MDIC

To improve standardisation of applying the RE-AIM framework for grant applications Kessler et. al., [26] developed the RE-AIM Model Dimension Items Checklist (MDIC), this sub-divides the domains into 31 criteria scored based on reporting within the publication. Prior to assessment against the RE-AIM MDIC [26] both reviewers (TH and AB) discussed the 31 criteria within the checklist to reach an agreement on the specific meaning of each criterion in the context of injury prevention trials in the studied population. This step was taken to ensure that each reviewer was clear on the fulfilment of each criterion of the RE-AIM MDIC within this context. The RE-AIM MDIC with definitions of each item as it relates to the context of sports injury prevention is provided in Table 1.

Each trial, plus any additional publications which reported information related to the original trial, were assessed independently by both reviewers (TH and AB) against the RE-AIM MDIC. Following independent assessment, a consensus was reached between both reviewers (TH and AB) with a third reviewer (JO) for conflict resolution.

**Table 1. RE-AIM framework MDIC criteria (adapted for sports injury context) with percentage of studies reporting each item on the checklist.**

| RE-AIM dimension and criteria | | | | | | Inclusion percentage |
|---|---|---|---|---|---|---|
| **Reach** | | | | | | |
| Exclusion Criteria (% excluded or characteristics of those excluded) | | | | | | 47 |
| Proportion of individuals who participated based on valid denominator | | | | | | 56 |
| (Valid denominator defined as approximate target population, e.g., number of players, schools in region) | | | | | | |
| Characteristics of participants compared to non-participants or to target population | | | | | | 18 |
| Use of qualitative methods to understand reach and/or recruitment | | | | | | 33 |
| **Effectiveness** | | | | | | |
| Measure of primary outcome–injury incidence | | | | | | 100 |
| Measure of broader outcomes–e.g., risk factors, performance metrics | | | | | | 36 |
| Sub-group analysis–Analysis WITHIN intervention group for differential responses to intervention | | | | | | 69 |
| Measure of short-term attrition (%) AND specifying treatment group i.e., intervention or control | | | | | | 60 |
| Use of qualitative methods to understand effectiveness | | | | | | 38 |
| **Adoption–Setting Level** | | | | | | |
| Settings excluded (% or reasons for exclusion) | | | | | | 51 |
| Proportion of settings accepting participations/refusing participation | | | | | | 73 |
| Characteristics of settings participating compared to non-participating/refused | | | | | | 3 |
| (Either settings refusing to take part or alternative settings in the target population area) | | | | | | |
| Use of qualitative methods to understand setting level adoption | | | | | | 35 |
| **Adoption–Delivery agent level** | | | | | | |
| Delivery agents excluded (% excluded or characteristics of those excluded) | | | | | | 5 |
| Proportion of delivery agents accepting participation/refusing participation | | | | | | 17 |
| Characteristics of delivery agents participating compared to non-participating/refused | | | | | | 0 |
| Use of qualitative methods to understand delivery agent level adoption | | | | | | 26 |
| **Implementation** | | | | | | |
| Measure of consistency of delivery e.g., compliance/adherence to the intervention reported | | | | | | 69 |
| Adaptions made to the intervention reported | | | | | | 16 |
| Cost of the intervention–either time or money reported | | | | | | 87 |
| Compliance/adherence across sub-groups/control or intervention reported | | | | | | 31 |
| Use of qualitative methods to understand implementation | | | | | | 40 |
| **Maintenance individual level** | | | | | | |
| Primary outcome (measure of injury incidence) reported at least 6 months following intervention period? | 4 | | | | | |
| Measure of broader outcomes reported | | | | | | 2 |
| Sub-group analysis of response of subgroups reported | | | | | | 4 |
| Long term attrition at 6 months (%) AND specifying treatment group | | | | | | 13 |
| Use of qualitative methods to understand individual maintenance | | | | | | 11 |
| **Maintenance setting level** | | | | | | |
| Intervention still on-going at 6 months reported | | | | | | 7 |
| Intervention has been adapted or elements retained reported | | | | | | 4 |
| Intervention has been incorporated into policy/curriculum reported | | | | | | 2 |
| Use of qualitative methods to understand setting level maintenance | | | | | | 11 |

Each criterion within the checklist was coded as either 'yes' if the criterion had been fully reported, 'no' if the criterion had not been reported, 'yes–inappropriate' if the criterion was reported either incompletely or inappropriately or 'not applicable' if the criterion did not apply to that study. Each study was given a percentage of reporting calculated by the equation ('yes'/total applicable) x100. This methodology aligns with previously published literature utilising the RE-AIM MDIC in this context [22, 23]. The level of reporting for each of the five

domains of the RE-AIM framework (reach, effectiveness, adoption, implementation, maintenance) was given as the mean score across the individual RE-AIM MDIC items within each domain. The designation of 'not-applicable' was used when it was deemed inappropriate to penalise a lack of reporting on elements which were not part of the study by design. For example, when programmes were delivered by the research team studies were not penalised for criteria within the *'adoption–delivery agent level'* domain as no delivery agents were involved in the study. In these cases, reported percentages of RE-AIM MDIC components are for all eligible trials.

Frequency analysis was conducted to produce level of reporting of the RE-AIM MDIC criteria, the data was analysed for relationships between year of publication and RE-AIM MDIC score. Tests of normality indicated a normal distribution for the MDIC percentage (Shapiro-Wilk test result $W(40) = 0.951$, $p = 0.079$) and a non-normal distribution for year of publication (Shapiro-Wilk test result $W(40) = 0.903$, $p = 0.002$). Therefore, Spearman's rank correlation coefficient was used to calculate the strength of association.

All data was imported into SPSS™ statistics for Mac, version 27 (IBM Corp., Armon, N.Y., USA) for quantitative analysis and statistical testing.

### Assessment with Consensus on Exercise Reporting Template (CERT)

All trials were assessed independently by two reviewers (TH and AB) against the Consensus on Exercise Reporting Template (CERT) 16-item checklist [31]. Following independent assessment consensus between the reviewers (TH and AB) was reached with any disputes referred to a third reviewer (JO) for resolution. The maximum possible score attainable is 19 points, with criteria covering elements of exercise interventions in the categories of: materials, provider, delivery, location, dosage, tailoring and planning/execution. To fulfil the criteria the study was required to provide details within the publication, supplementary material or provide a direct link to online resources. The criteria were deemed to have not been fulfilled if a study provided a reference to another study utilising the same programme. Scores were reported as a percentage of fulfilled criteria on all applicable elements, any elements that did not apply to the study were designated 'not-applicable' and not included in the score percentage. For example, the description of home-based exercise components if the study was not home based. The CERT proforma used can be found in the S2 Table.

## Results

### Search strategy

The search strategy identified a total of 3306 articles, once duplicates were removed. Title and abstract screening excluded 3131 articles, leaving 175 articles sought for retrieval. Of these 175, five were not available in the English language, leaving 170 articles that had their full text screened against the eligibility criteria. A lack of injury incidence data resulted in 94 articles being removed, 27 were removed due to inclusion of participants aged 19 and older, three articles used non-exercise-based interventions and one article was a secondary analysis of previously published data. Four articles were identified as follow-up papers relating to the original trials that would be assessed in this review and so were retained. This process left 41 articles reporting on unique trials that met the eligibility criteria, with an additional 18 articles that reported on data pertaining to those trials. Updated searches performed in May 2023 identified a further four articles reporting on unique trials and an additional two follow-up papers reporting on previous trials (Fig 1).

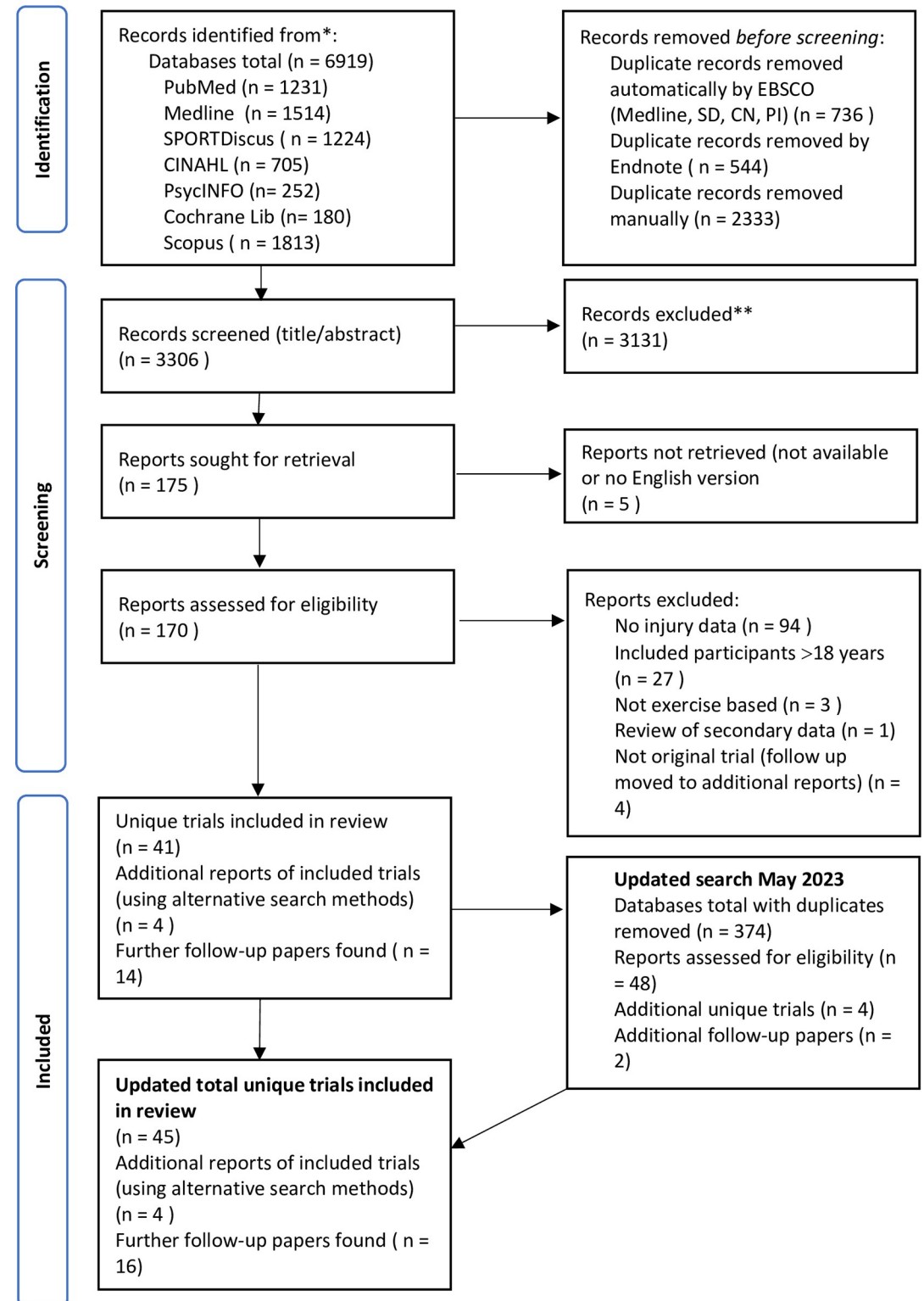

**Fig 1. Preferred Reporting Items for Systematic Reviews and Meta-Analyses (PRISMA) flowchart [34] detailing search results.**

## Study characteristics

Details of the reviewed studies can be found in S3 Table. The earliest study was published in 1999 and the most recent in 2023. Thirty-three of the 45 studies were of randomised control trial (RCT) design with 12 either quasi-experimental or a cohort study with a control/comparator group. Nineteen studies included both sexes, with 13 including only female participants and the remaining 13 including only male participants. The smallest sample size was 22 participants, with the largest containing 4564 participants. Participants ranged from as young as seven to eighteen years of age (mean age 13.1 years, SD ± 2.3), studies were excluded if they contained any participants aged 19 years and older. The most popular sport studied was soccer (n = 25) followed by basketball (n = 8) and handball (n = 5). Twenty-seven studies were conducted exclusively within a sports club setting, 17 were conducted exclusively within a school setting and one was conducted within both a school and sports club setting [13]. Studies were conducted across 16 different countries. The largest number of studies were conducted in the United States of America (n = 8), followed by Canada (n = 7), Japan (n = 4), Sweden (n = 4) and Norway (n = 3) and The Netherlands (n = 3). Australia, Denmark, Germany, Iran and The United Kingdom were the origin of two studies each. Chile, Finland, Greece, Saudi Arabia and Tunisia contributed the sample to a single study each. One study [15] was conducted across four countries, The Czech Republic, Germany, The Netherlands and Switzerland.

The majority of the studied interventions were classified as multimodal neuromuscular training (n = 34), followed by balance training (n = 5), muscle strengthening (n = 2) and combined stretching and strengthening (n = 2). One study [48] studied the impact of landing exercises in Australian football, whilst Azuma and colleagues [49] investigated static stretching for injury prevention in soccer.

The intervention lengths ranged from 5 minutes [50] to 90 minutes [51]. The mean intervention length was 22 minutes with a median of 20 minutes. The length of intervention study periods ranged from six weeks [52] to one and half years [50] the mean length was 30 weeks with a median length of 28 weeks. The mean intervention study period for RCTs was 27 weeks, with a median of 24 weeks. The mean intervention study period of non-randomised trial designs was 34 weeks with a median of 42 weeks.

## Risk of bias assessment

Risk of bias assessment for RCTs was conducted using the RoB 2 tool [46]. Of the 33 RCTs in this review twelve were designated as 'low-risk' for bias, 17 were designated as having 'some-concerns' for bias and four were designated 'high-risk' for bias. The most common reason for a designation of 'some-concerns' for bias was a lack of trial registration and/or an explicit analysis plan within the registration (22 of 33 trials failed to report trial registration or an analysis plan). An example of a commonly used trial registration database is the 'International Standard Registered Clinical/soCial sTudy Number' registry, which changed its name from the 'International Standard Randomised Control Trial Number' to allow the inclusion of non-randomised trials [53]. Authors are encouraged to register all details of the study protocol including primary and secondary outcomes as well as a comprehensive data analysis plan. Trial pre-registration is also recommended within the Consolidated Standards of Reporting Trials (CONSORT) statement [54], which is a 25-item checklist designed at improving the standardisation and quality of trial reporting in randomised control trials. A lack of data on compliance to the intervention (seven trials) led to a designation of 'some-concerns' due to potential deviations from the intended intervention or a lack of blinding of the outcome assessors (nine trials). The designation of 'high-risk' of bias was applied in four trials due to a lack of data on participant numbers included in the final analysis and/or dropout rates.

Non-randomised trials in this review were assessed for risk of bias using the ROBINS-I tool [47]. Seven of the 12 studies were designated 'moderate' risk of bias, with the remaining five designated 'Serious' risk of bias. A lack of trial registration was the predominant reason for a designation of 'moderate' risk of bias, with all studies failing to register the study protocol or analysis plan in advance. Whilst it has historically been RCTs which have required registration [55], databases such as ISRCTN now accept registrations for non-randomised trials, and guidance from the Cochrane Library [47] strongly encourages it. A designation of 'serious' risk of bias was applied due to a lack of data on dropout/number of participants in final analysis (five trials) and concerns over selection into the study (two trials). A full transcript of risk of bias assessment for each study is provided in the S4 and S5 Tables.

## Reporting assessment with the RE-AIM MDIC

The mean RE-AIM MDIC score across all trials was 31% (SD ± 16.2%, Range 7–77%). The domain 'effectiveness' was the most comprehensively reported with a mean of 60% followed by 'implementation' with 48%, 'adoption–setting level' with 40%, 'reach' with 38%, 'adoption–delivery agent level' with 12%', 'maintenance–individual level' with 7% and 'maintenance–setting level' with 6%.

The mean RE-AIM MDIC score for RCTs was 33% (SD ± 14.3, Range 7–58), with non-randomised study designs having a mean of 26% (SD ± 20.0, Range 7–77%) (Table 2).

The mean RE-AIM MDIC score for sports club-based interventions was 30% (SD ± 14.5, Range 7–58%) and for school-based interventions the mean was 34% (SD ± 19.1, Range 7–77%) (Table 3). Mean reporting frequency across all studies for each of the 31 criteria of the RE-AIM framework MDIC is shown in Fig 2.

## Reporting of RE-AIM dimensions

**Reach.**   The mean level of reporting across all four 'reach' components (**MDIC items 1–4**) was 38% (range 18–56%). Either the percentage of excluded participants OR explicit exclusion criteria (**MDIC item 1**) was reported in 21 (47%) trials, with the number of participants taking part relative to a valid denominator i.e., appropriate calculation of target population, (**MDIC item 2**) reported in 25 (56%) trials. A comparison of participants with non-participants in the target population (**MDIC item 3**) was reported in eight (18%) trials. Qualitative methods to further understand reach (**MDIC item 4**) were reported in 15 (33%) trials.

**Effectiveness.**   The mean level of reporting across all five 'effectiveness' components (**MDIC items 5–9**) was 60% (range 36–100%). All 45 trials reported the primary outcome measure (**MDIC item 5**), which was a measure of sports injury incidence. Sixteen (36%) trials reported broader outcomes in addition to the primary outcome (**MDIC item 6**). Sub-group analysis (**MDIC item 7**) was conducted in 31 (69%) trials and was predominantly comparisons of differential injury rates between sexes in the intervention group. Short term rates of attrition (**MDIC item 8**) were reported in 27 (60%) trials, to meet this criterion the group from which

**Table 2. RE-AIM MDIC score by domain and CERT for all studies, randomised controlled trials and non-randomised studies.**

| Study design | Reach | Effectiveness | Adoption—Setting | Adoption—Delivery agent | Implementation | Maintenance-Individual | Maintenance—Setting | Overall mean MDIC score | CERT Score |
|---|---|---|---|---|---|---|---|---|---|
| All studies (n = 45) | 38% | 60% | 40% | 12% | 48% | 7% | 6% | 31% | 50% |
| Randomised controlled trial (n = 33) | 42% | 65% | 44% | 12% | 54% | 4% | 5% | 33% | 54% |
| Non-randomised design (n = 12) | 27% | 48% | 30% | 13% | 35% | 13% | 8% | 26% | 39% |

**Table 3. RE-AIM MDIC score by domain and CERT for all studies, club-based studies and school-based studies.**

| Study setting | Reach | Effectiveness | Adoption—Setting | Adoption—Delivery agent | Implementation | Maintenance -Individual | Maintenance—Setting | Overall mean MDIC score | CERT Score |
|---|---|---|---|---|---|---|---|---|---|
| All studies (n = 45) | 38% | 60% | 40% | 12% | 48% | 7% | 6% | 31% | 50% |
| Club (n = 28)* | 34% | 58% | 42% | 11% | 48% | 7% | 5% | 30% | 48% |
| School (n = 18)* | 46% | 62% | 41% | 14% | 49% | 8% | 7% | 34% | 53% |

*note—Emery et al., 2022 was conducted in both a school and club setting so was included in both analyses

the attrition occurred i.e., intervention or control, had to be specified. Qualitative methods to further understand effectiveness (**MDIC item 9**) were reported in 17 (38%) trials.

**Adoption–setting level.** The mean level of reporting across the four components of 'adoption—setting level' (**MDIC items 10–13**) was 40% (range 3–70%). The proportion of potential settings excluded by the trial researchers or the rationale for exclusion (**MDIC item 10**) was reported in 19 (47% of eligible) trials. The proportion of settings accepting the invitation to

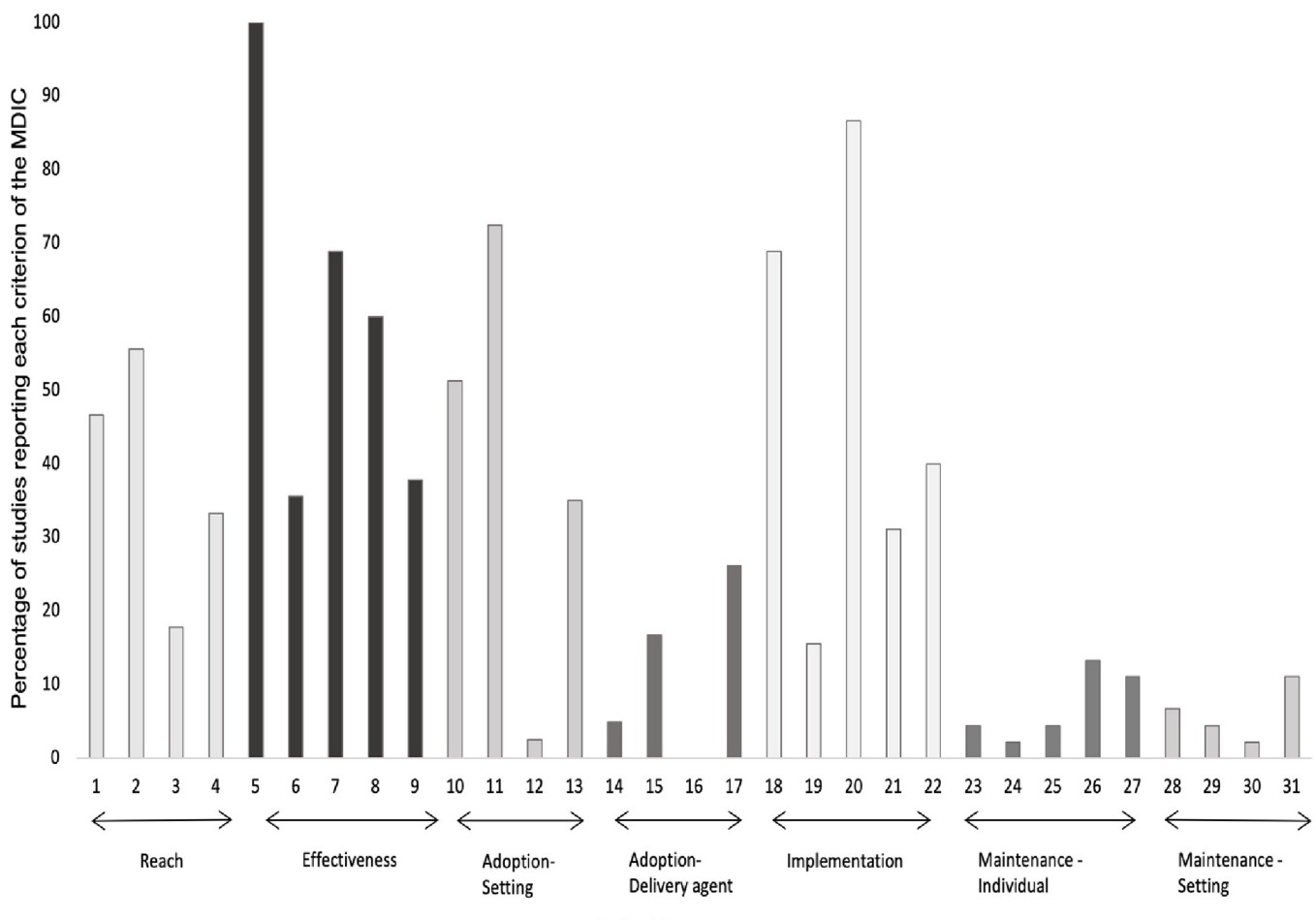

**Fig 2. The percentage of trials reporting each of the 31 criteria of the RE-AIM model dimension checklist (MDIC) [23].**

take part in the research (*MDIC item 11*) was reported in 28 (70% of eligible) trials. A comparison of characteristics between the settings agreeing to participate and those who declined (*MDIC item 12*) was reported in one trial [50]. The use of qualitative methods to understand why settings accepted or declined participation (*MDIC item 13*) was reported in 15 (38% of eligible) trials.

**Adoption–delivery agent.**   The mean reporting across the four components of '*adoption —delivery agent level*' (*MDIC items 14–17*) was 12% (range 0–26%). The percentage of delivery agents excluded by trial researchers (*MDIC item 14*) was reported in one trial [56]. The proportion of delivery agents agreeing to take part (*MDIC item 15*) was reported in seven (17% of eligible) trials and zero trials reported any comparison between delivery agents agreeing to take part and those two declined (*MDIC item 16*). Qualitative methods to further understand the adoption by delivery agents (*MDIC item 17*) were reported in ten (24% of eligible) trials.

**Implementation.**   The mean level of reporting across the five components of '*implementation*' (*MDIC items 18–22*) was 48% (range 16–87%). Reporting of a measure of consistency of delivery (*MDIC item 18*) was found in 31 (69%) trials, this was typically compliance with the intervention in terms of sessions completed across the whole study period and average sessions per week. Adaptations made to the programme (*MDIC item 19*) were reported in seven (16%) of trials. Reporting the cost of the intervention, either time or monetary (*MDIC item 20*) was found in 39 (87%) of trials. The consistency of delivery across different sub-groups (*MDIC item 21*) was reported in 14 (31%) trials. Qualitative data to further understand implementation (*MDIC item 22*) was reported in 18 (40%) of trials.

**Maintenance–individual.**   Mean reporting across the five components of '*maintenance— individual level*' (*MDIC items 23–27*) was 7% (range 2–13%). Reporting of the primary outcome, injury incidence, six months or greater after the intervention period was finished (*MDIC item 23*) was found in two (4%) trials [17, 57], with long-term broader outcomes (*MDIC item 24*) reported in one (2%) trial [16]. Long-term sub-group analysis (*MDIC item 25*) was reported in two (4%) trials [13, 57].

Six trials (13%) reported a measure of long-term attrition (*MDIC item 26*) and five (11%) reported qualitative measures to support individual maintenance (*MDIC item 27*).

**Maintenance–setting.**   Average reporting across the four components of '*maintenance— setting level*' (*MDIC items 28–31*) was 6% (range 2–11%). Three trials (7%) [57–59] reported whether the intervention was still on-going at least six months after the completed study period (*MDIC item 28*). Two trials (4%) reported that adaptations had been made to the intervention post trial (*MDIC item 29*) [16, 57], this was either proportion of exercises that were retained [16, 57] or how the intervention was used in sports other than what it was designed for [57] Intervention integration into policy/curriculum (*MDIC item 30*) was specified in one trial [58] and qualitative data to further understand setting maintenance (*MDIC item 31*) was reported in five (11%) trials.

## RE-AIM MDIC score for publication year

When subjected to bivariate analysis using Spearman's rank correlation coefficient, there was a moderate positive correlation ($r(43) = .480$, $p < .001$) between publication year and RE-AIM MDIC score. This demonstrated that RE-AIM MDIC scores are improving with more recent publications.

## Consensus on Exercise Reporting Template (CERT)

Reporting of elements pertaining to the intervention investigated by the trials were appraised against the CERT [29] checklist. The mean reporting score across all studies was 50%

(SD ± 20.8, range 0–81%). Analysis of correlation between year of publication and CERT score revealed a weak positive correlation ($r(43) = 0.270$, $p = 0.073$), demonstrating a small improvement in CERT scores with more recent publications.

## Discussion

The aim of this review was to assess the reporting of factors related to the impact of injury prevention trials in child and adolescent sports using the RE-AIM framework [25] and CERT [31]. This review only evaluated studies on child and adolescent populations as impact factors may differ in comparison to adult populations, such as the increased reliance on programme deliverers [60] and involvement of parents [61]. In addition, sport in younger populations takes place in both schools and sports clubs, which may present their own challenges for implementation of sports injury prevention programmes. Studies included in this review were found to focus on reporting details contained within the '*effectiveness*' domain of the RE-AIM framework, whereas information relating to '*implementation*' and '*reach*' were less well reported. Details concerning '*adoption*' and '*maintenance*' were frequently neglected, which is concerning, considering how critical adoption and long-term maintenance are to the impact of a health intervention [62]. This supports the findings of previous reviews applying the RE-AIM framework to sports injury prevention literature [22, 23]. A distortion towards reporting data related to the efficacy or effectiveness of an intervention, whilst neglecting other areas of implementation was also found in other areas of health intervention literature [29, 30], indicating that the weaknesses identified in reporting are not confined to sports injury prevention research [63].

The gap between science and its translation into practice has previously been highlighted in the field of sports injury prevention [39], with more recent literature suggesting that little has changed over the past decade [64]. To address this gap, there is a need to incorporate implementation science within sports injury prevention research [39] and identify weaknesses in trial design and reporting that are contributing to the gap. Addressing the disparity between research on intervention efficacy and real-world results was the primary aim behind the creation of the RE-AIM framework [25]. Evaluation of the RE-AIM framework 20 years after its inception suggests that it has become widely used in public health research as a tool for intervention planning and evaluation [62] and this appears to have positively impacted the impact of health-related programmes in the community [65].

The RE-AIM framework defines the domain of '*reach*' as how well the participants in the study reflect the characteristics and specific requirements of the target population [25]. The representativeness of the participant cohort to the target population will dictate the utility and impact of an intervention, key components of external validity [66]. For example, both the efficacy study [12] and effectiveness study [57], on the rugby specific injury prevention programme '*Activate*', were only conducted in boys. The effectiveness of the intervention in reducing injures in boys supports its use in schoolboy rugby, but it is not clear if it will be effective for girls rugby, a sport which is increasing in popularity in the United Kingdom [67]. This example illustrates the need for studies to report on how their participants represent the target population and whether the intervention is likely to address the needs of the wider community. Only eight of 45 studies in this review drew any comparisons between the participants in the study and the target population, this lack of comparison reduces the studies external validity and makes it unclear whether the intervention will address the needs of the wider population.

These findings contrast broader public health intervention research where almost two thirds of studies reported comparisons between the participants and the target population [30]

and in general scoring across the *'reach'* domain was markedly higher (54% [30] vs 38% in the current review). This is most likely due to the requirement for public health interventions to address the needs of the target population in order to justify their cost.

These considerations are still applicable in the context of sports injury prevention where institutions, clubs and schools will need to decide which intervention will best suit the needs of their population whilst considering the available resources. Future research in this field should provide more clarity on the needs of the target population, the representativeness of their study sample and how the results of their study will benefit the wider community.

As previously identified, the level of reporting for components contained within the *'effectiveness'* domain was far higher than other domains within RE-AIM framework, with an average across the five components of 60% compared with the next most frequently reported domain of *'implementation'* with 48%. The terms *effectiveness* and *efficacy* often refer to the conditions under which the trial was conducted, with an efficacy study conducted under ideal circumstances and an effectiveness study conducted in real-world conditions [34]. However, these definitions may not describe the true difference between efficacy and effectiveness studies, often leading to effectiveness being used to describe a trial conducted in the 'real-world' that otherwise does not satisfy the criteria for effectiveness [68]. These criteria include whether the populations were in their habitual setting, whether they represented the diversity within the population and whether the sample size and study duration were adequate to see meaningful effects [68]. It has also been suggested that to qualify as studying effectiveness a trial must consider implementation and ideally utilise frameworks such as RE-AIM to ensure that the study outcomes are translatable to practice [69]. Rather than two distinct study types it is more likely that efficacy and effectiveness exist on spectrum, increased consideration of elements that relate to real-world conditions move the needle towards effectiveness and make it more likely that an intervention will have a beneficial and long-lasting impact [68].

Although it would appear that moving towards effectiveness from efficacy is beneficial it can present a challenge with intervention research, which is managing the potential conflict between internal and external validity [34]. Randomised controlled designs tend to favour the promotion of high internal validity, conferring greater trust in the results, whilst compromising on external validity, how the results impact the wider population [70]. RCTs demonstrated considerably higher reporting levels for the *'efficacy/effectiveness'* RE-AIM domain in comparison to non-randomised designs (65% vs 48%), suggesting a link between components of this domain and measures of internal validity.

To support the primary outcome, the RE-AIM framework encourages the reporting of data which help explain *why* the intervention may have demonstrated effectiveness. One third of trials supported injury incidence data with secondary outcomes such as impact of the intervention on balance metrics [71], measures of strength [72] and flexibility [73]. These additional data points may help identify the mechanism by which the intervention was effective and potentially indicate possible risk factors for injury, supporting the utility of the intervention and identification of the target population. Two thirds of studies incorporated sub-group analysis such as sex, level of compliance, or intervention dose. Sub-group analysis based on compliance is particularly relevant as it has consistently been shown that high compliance leads to greater effectiveness [19, 41]. Understanding the relationship between compliance and effectiveness has implications for recommended intervention dose and frequency. If an intervention requires a high level of compliance and/or frequency to be effective it may not be suitable in settings that cannot deliver this dose, requiring the need to explore more feasible options. Sex sub-group analysis was frequently reported and in some instances the intervention was only effective in one of the sexes [59, 74], this links back to the domain of *'reach'* and whether the studied intervention will work for the intended population. Short-term attrition rates,

especially when supported with reasons for the attrition, can be an indicator of the acceptability of the intervention [75]. Just over half of the studies reported the short-term attrition rate, including whether it occurred in the control or intervention group, this has implications for acceptability but also may impact the validity of the data analysis [76]. The lack of reporting attrition rates was a common reason for raising risk of bias concerns due to the implications for internal validity of the study results.

Each domain of the RE-AIM framework contains a criterion for qualitative data reporting relating to that domain, a useful source of supporting information that is frequently overlooked in quantitative studies [66]. When qualitative data was reported it was often focussed on the negatives associated with the intervention, such as difficulties in collecting data and reasons for attrition. However, the use of post-trial questionnaires administered to coaching staff in the study by Rössler et al., [15] provided useful information regarding the attitudes of coaches towards the intervention. This evaluation revealed that coaches felt that injury prevention was a high priority and that the programme could prevent injuries and improve performance [15]. Concerningly, coaches rated the time requirement as 'only just reasonable' indicating that the 20-minute injury prevention warm-up may be at risk if time becomes a greater pressure. Taking this a step further Barden and colleagues conducted a series of semi-structured interviews with rugby coaches in conjunction with the effectiveness study on the schoolboy rugby intervention 'Activate' [60]. They found that the pupils themselves showed little awareness of 'Activate' and therefore there was an increased reliance on coaches for intervention delivery [60]. Coaches reported adapting the programme based on their specific requirements as well as utilising the programme in different sports [60]. These findings are positive for the potential of long-term programme adoption and maintenance as they demonstrate a willingness for programme deliverers to incorporate the intervention into their own practice. However, a lack of awareness from the pupils themselves may limit the degree to which they can act as programme delivers and questions whether they can conduct the intervention correctly without supervision [60].

Whilst it may be beyond the scope of a study to conduct full-length interviews with programme deliverers of the intervention, the post study questionnaires as used by Rössler et al., [15] are relatively simple to conduct and can give a wealth of information regarding implementation and potential barriers to programme adoption and maintenance. This process of evaluation should be considered normal practice in the development of an intervention, especially where novelty exists in the design or target population, creating an iterative cyclical process of '(re) evaluate, identify and intervene' as stipulated in the Team Sport Injury Prevention (TIP) cycle [77].

Across the studies in this review data within the domain of 'implementation' was reasonably well reported, with 87% of studies reporting the time or monetary cost of the intervention and two thirds reporting a measure of consistency of delivery such as compliance. Compliance or adherence has been demonstrated to significantly impact the effectiveness of an injury intervention programme, with greater compliance associated with lower risk of injury [41]. Compliance, along with attrition, also indicates the acceptability of an intervention and a low compliance rate should raise red flags that require further investigation [75]. Despite its potential importance for policy decision making only one trial attempted to provide a cost for the delivery of the intervention in monetary terms [56]. Many trials reported time costs for delivering the intervention, an important consideration for acceptability as an intervention that is too long was identified as a significant barrier to implementation in schools [78]. A lack of detail regarding the implementation of exercise programmes has been identified as a weakness of the RE-AIM framework in the context of sports injury prevention studies [23], where specific details of the exercise interventions are required for replication. The recently developed

CERT [31] was used in this review to compliment the RE-AIM framework and provide the increased level of detail required to scrutinise the studies on their reporting of key elements of the exercise intervention. The range of CERT scores was surprising, from 0% to 81% of criteria fulfilled, with some publications including photographs and links to exercise videos from electronic copies of the article. This level of detail provides both transparency of implementation and high reliability of successful replication and should become the standard in the digital age. Consistent weaknesses noted from the CERT analysis were the lack of rationale for exercise progressions and whether the fidelity of the exercise intervention was being measured and how. Whilst programme compliance provides useful information, it does not give a complete picture when considered without fidelity, i.e., the extent to which an intervention is delivered as planned [79]. Fidelity is especially important in exercise interventions, as small changes in which exercises are delivered and how could have consequences to the intervention's effectiveness [63]. Considering that many of the interventions in the reviewed studies were of significant length (median reported length 20 minutes) and often consisted of multiple exercises, providing data on fidelity, particularly relating to which session components were typically included or excluded, would seem to be essential in understanding effectiveness. Only three studies satisfied the fidelity criteria [80] (original trial by Slauterbeck et al., [45]), [57, 81] (original trial by Emery et al., [74]) and given the results from the fidelity study conducted by Krug et al., [80] found only 19% of the exercises were being included and only 66% being performed with 'good' technique, there is clearly a greater need for inclusion of fidelity analysis in future studies.

Even if an intervention can demonstrate effectiveness and has been successfully implemented as part of the research study, it will have limited impact if it is not adopted by programme delivers and the settings where it will be carried out. Within the RE-AIM framework, 'adoption' is broken into 'delivery agent level' and 'setting level', allowing the differentiation of factors which may not apply to both levels. Reporting across both levels of 'adoption' was found to be considerably lower than wider health intervention research [30], with data on 'adoption—delivery agent level' particularly poor (11% mean reporting vs 18% in Gaglio et al., [30]). Studies frequently reported on acceptance or rejection for taking part in the study by the setting, at times including a rationale for rejection such as a lack of time available [50, 74], regarding injury prevention as a low priority [82] or the inability to provide adequate exposure and injury data [83]. Rarely were the delivery agents themselves addressed, with only one study [56] reporting how many had accepted the invitation to take part. When reported, qualitative data in the 'adoption' domain can be enlightening, for example, Soligard and colleagues commented that many of the coaches in community sport were parents or volunteers [18]. This information has implications for both time available and the expertise required to deliver an injury prevention intervention. Within child and adolescent sport, a greater focus needs to be placed on those delivering the intervention, as the level of autonomy from the athletes themselves is potentially lower and greater expectation is placed on the coaching staff for delivery of session content [84].

The RE-AIM framework divides the 'maintenance' domain into 'individual level' and 'setting level'. 'Maintenance—individual level' refers to data pertaining to the study participants, rather than the actions of the participants, for example, does a study report data such as the primary outcome at least six months after the intervention has finished, as opposed to whether the individual was still performing the intervention. Only two studies [17, 57] reported injury data at least six months post intervention period. In the study by Mandelbaum and colleagues [17] this involved a season long observation study following the intervention period allowing the differentiation of injury rates between the intervention period and post intervention period. The study by Barden et al., [57] utilised a different approach, their study period took

place over three years after the efficacy study on *'Activate'* [12] was conducted. Following the efficacy study, *'Activate'* was disseminated by the Rugby Football Union (RFU) allowing Barden et al., [57] to compare injury rates for settings who had adopted the programme against those who had not. This demonstrated the shift in research focus as part of the research cycle, from efficacy of the intervention, where high internal validity was the aim, to study of adoption and dissemination, where external validity became more important. Although this methodology employed by Barden and colleagues [57] did allow a measure of the effectiveness of the intervention when the research team were no longer directly involved there are concerns over whether the programme fidelity was preserved. This was demonstrated in the variability in number of reported exercises included in each training session (4–15 exercises included) [57], although this does show how settings and deliverers may need to adapt a programme to suit their own requirements and limitations. Observing the impact of an intervention beyond the period of direct involvement by the research team provides important information on real-world effectiveness as well as measures of adoption and long-term maintenance. Given the seasonal nature of most sports an observation period lasting at least one season following the cessation of the study would provide valuable information on the true impact of the intervention and indicate if it is likely to be effective in the long-term.

*'Maintenance—Setting level'* refers to whether the intervention has been continued in the post-trial period, whether it has been incorporated into policy or curriculum and if it has been altered or adapted in any way. This domain was very poorly reported, with only three studies [57, 59, 85] reporting if the intervention was still being used six months or greater after the study period had ended and only one study reported whether it had become included in any policies, which it had not [85]. The paper by Lindblom et al., [85] was a follow up to the original trail by Waldén et al., [58] and was a comprehensive analysis of implementation conducted three years post-trial using the RE-AIM framework. They reported a high level of programme maintenance, with 82% of coaches from the trial still using the programme, however they also reported that despite its effectiveness the programme had not become formal policy [85]. The lack of translation into policy is a concern for long-term maintenance, as without formal policy the continuation of a programme will rely on knowledge and practice transfer between programme deliverers [86]. The need for multi-level support for the implementation of an injury prevention programme is emphasised in the RE-AIM Sports Setting Matrix (RE-AIM SSM) [87]. This extension of the RE-AIM framework is designed to expand the RE-AIM framework to encompass the different levels of sports delivery, including national level, regional level and club level [87]. Finch and Donaldson argue that at each level different factors influence successful implementation therefore researchers need to think more broadly when considering how an injury prevention programme can become sustainable.

Reporting levels across the whole *'maintenance'* domain fell short of those found in wider health research, with reporting levels at 7% for *'maintenance–Individual level'* and 6% for *'maintenance–setting level'*, compared to 34% and 18% respectively in health research [30]. The reasons for this disparity are likely multifactorial, such as the increased need to monitor longer-term adverse events in health trials, however it is possible that the publication pressure, which promotes reporting of short-term measurable outcomes in sports injury studies, exacerbates this problem [88]. It is also possible that data on programme maintenance has yet to be published, therefore the aim of this review is not to penalise individual studies, especially those more recently published, for not including maintenance data. However, it must be noted that no studies were found to have indicated an intention to collect and publish maintenance data at a later date.

If we are to fully understand how intervention programmes function in the long-term and facilitate their inclusion into policies, we need to consider how trial design and subsequent

publication supports this. A decade ago, Donaldson and Finch [39] suggested researchers needed to think more broadly, including implementation science in trial design and understanding the different levels of implementation. Whilst this does not appear to be have been universally adopted the approach taken by Waldén et al., [58] and Barden et al., [57] should serve as positive examples. In the example of Waldén et al., [58], the original trial was supported by two additional follow-up publications, an in-depth analysis of the influence of compliance on injury rates [89] and a three year follow up on implementation assessed against the RE-AIM framework [85]. A further publication reported the effectiveness of the intervention nationwide, following its uptake across Sweden in the time period since the original trial [90]. This type of approach shows that it is possible to incorporate both short and long-term data collection methods into a trial design and emphasises the role of the RE-AIM framework as a tool for post-trial analysis of impact. An alternative method is to utilise the RE-AIM framework prior to commencement of the trial, using it to inform the trial design. This was the approach taken by Barden and colleagues; the effectiveness study [57] built on the findings from the efficacy study [12] by utilising the RE-AIM framework to inform the implementation of the intervention [28]. Barden and colleagues explored the attitudes of coaches to injury prevention in rugby to ascertain barriers and facilitators to the intervention [60] and utilised behaviour change models to analyse the findings of surveys taken from coaching workshops [91]. The result was a study which can be regarded as studying real-world effectiveness; *'Activate'* adopters had a 23% lower match injury rate and a 59% lower training injury rate [57]. This supported the results from the efficacy study [12] and provides powerful evidence that *'Activate'* works in schoolboy rugby *if* it is adopted. The next step in this process would be to further understand the rationale for non-adoption and what further barriers have to be overcome.

Sub-group analysis of RE-AIM MDIC scores produced some interesting patterns, especially when comparing randomised controlled trial (RCT) designs to non-randomised designs. Overall RE-AIM MDIC scores were higher in RCT study designs compared with non-randomised designs (33% vs 26%), however when individual domains were analysed RCTs scored higher on *'reach'*, *'effectiveness'*, *'adoption'* and *'implementation'* but considerably lower on *'maintenance'*. This is likely a consequence of the constraints that come with the RCT design, often limited on study duration (in this review the median length of intervention for RCTs was 24 weeks vs 42 weeks for non-randomised design) due to study funding and risk of contamination with prolonged study periods [92]. The inclusivity and consequential diversity of participants in non-randomised designs often leads to higher external validity, meaning that outcomes are more likely to translate outside of the study research setting [92]. One of the main drawbacks of non-randomised designs is the higher risk of selection bias, often raising concerns over the internal validity of results [93]. Choosing between the promotion of internal or external validity is also a function of the intervention design process. Efficacy studies, aiming for high internal validity to establish causation, often precede implementation and feasibility studies, which favour external validity [37]. It is therefore important to consider that many of the studies assessed in this review may be in the early stages of intervention design, with subsequent implementation studies planned for the future. It has been argued that efficacy studies and implementation studies needn't be separate, with an ideal study design combining elements of both [40]. This approach is rare in sports prevention research [22], but has been proposed for the '*Prep-to-Play*' hybrid implementation-effectiveness RCT in women and girls Australian Football [94]. This method has the benefit of reducing the length of time required to develop and intervention and should speed the process of converting research into practice [40]. This method, whilst promising for sports injury prevention research does create the challenge of greater complexity of analysis and therefore should be approached with caution [38].

This review found that reporting details related to external validity, as measured by the RE-AIM MDIC is improving with more recent publications. This is also evident in the comparison between the results of this review and a review conducted almost 10 years ago in sports injury prevention [23]. Whilst the results *for 'reach'*, *'effectiveness'*, *'adoption'* and *'implementation'* were largely similar the current review found an improvement in both *'maintenance—individual level'* (7% vs 1% in O'Brien and Finch [23]) and *'maintenance–setting level'* (6% vs 0% [23]).

The examples of *'Activate'* [12, 28, 57, 60, 91] and the knee injury specific neuromuscular training programme in Sweden *'Knäkontroll'* [58, 85, 89, 90] have demonstrated how implementation science can be incorporated into research on sports injury prevention as part of a series of studies. One criticism of this approach would be the need to read multiple papers published over a number of years, in order to appreciate the full picture, making it difficult for end users who are not researchers to access and digest the information. The role of researchers is to not only design studies which reflect real-world conditions, but also ensure that the study results are communicated in a way that allows both researchers and practitioners to access and understand the findings.

## Limitations

This review has systematically appraised reporting levels of injury prevention trials in child and adolescent sport and builds on previous reviews utilising the RE-AIM framework by adding greater depth of analysis provided by the CERT. In doing so it has provided a novel insight into the reporting of impact factors in sports injury prevention studies and identified weaknesses, such as a focus on effectiveness, that need to be addressed to help close the gap between science and practice. However, several limitations have been identified which must be considered in interpretation of the findings. An extensive search strategy was used however it is still possible that relevant trials were missed due to computer error in searches, human error in screening or whilst removing duplicates. Whilst further searches were conducted to identify any additional papers pertaining to the original trial, such as follow-up studies or secondary analysis, it is possible that these were missed or have been published since the conclusion of searches.

The RE-AIM framework is a well-established framework that has been cited by over 2800 publications and has been used to inform and evaluate trial design in more than 450 studies [62]. However, the assessment of the RE-AIM MDIC has a degree of subjectivity as criteria are laid out as a guide without explicit instruction of what can or cannot be deemed to have satisfied the criteria. To improve the reporting quality, this review used a consensus between two reviewers following independent assessment of the trial. The CERT [31] has been created more recently than the RE-AIM framework, and as such does not have the volume of evidence supporting its utility. However, it appears to demonstrate good inter-rater agreement [33] and has been utilised in numerous systematic reviews despite its recent creation [32].

## Conclusion

Despite an increasing reporting frequency of RE-AIM elements within more recent research there is still a low level of reporting, with many studies not exceeding 50% of reporting of criteria contained within the RE-AIM MDIC. Reporting focussed on details within the *'effectiveness'* and *'implementation'* domains, often neglecting other areas such as *'reach'*, *'adoption'* and *'maintenance'*. Reporting details within *'reach'* is particularly important in gauging the studies external validity, whilst neglecting adoption and maintenance raises concerns over the long-term use and impact of the studied interventions. Reporting of the exercise intervention

details, assessed with the CERT, was equally low, with an average of 50% of items reported across all studies. An improvement in reporting levels contained within the *'maintenance'* domain of the RE-AIM framework in comparison to a previous review conducted almost a decade ago [23] suggests that there is now a greater consideration of the sustainability of an intervention, although this consideration is still rare.

The challenge of balancing internal and external validity in intervention studies often means a choice between a focus on efficacy or a focus on implementation and feasibility. Many of the studies in this review may have been in the early stages of the research cycle, with a focus on efficacy rather than translation of the research to practice. The translation to practice will only take place if the research cycle is continued to include implementation and feasibility studies, which is not always the case. Therefore, this multiple study, sequential approach may not be optimal, with the alternative hybrid '*effectiveness-implementation*' design offering a quicker and more effective translation of research to practice [40]. This review has highlighted that although the hybrid methodology design promoted by Curran et al., (2012) was proposed over a decade ago it has yet to be utilised in the field of sports injury research in child and adolescent populations. This should be about to change following the trial registration of the *'Prep-to-Play'* RCT for injury prevention in Australian Football which plans to utilise this method [94].

There is a wealth of information on the effectiveness of a range of different exercise-based injury prevention interventions in diverse populations, yet little knowledge about to implement them optimally. Therefore, this review recommends that future research should focus more on the translation of research to practice, conducting studies on the implementation and feasibility of interventions we already know are effective at reducing injury. It is also recommended that when novel interventions are trialled, a hybrid effectiveness-implementation design is employed. Utilising this method should reduce the time lag between developing an effective intervention and understanding how best to implement it. This should speed the translation of research to practice and provide more useful information for policy makers.

## Supporting information

**S1 Checklist. PRISMA 2020 for abstracts checklist.**
(DOCX)

**S2 Checklist. PRISMA 2020 checklist.**
(DOCX)

**S1 Table. Online search strategy.**
(DOCX)

**S2 Table. Consensus of reporting template (CERT).**
(DOCX)

**S3 Table. All reviewed studies and additional papers.**
(DOCX)

**S4 Table. Risk of bias assessment of randomised trials using the Rob 2 tool.**
(DOCX)

**S5 Table. Risk of bias assessment of non-randomised studies using the ROBINS-I.**
(DOCX)

## Author Contributions

**Conceptualization:** Thomas Hughes, John O'Hara, Nick London, Stacey Emmonds.

**Data curation:** Thomas Hughes, Alan Burton.

**Formal analysis:** Thomas Hughes, Alan Burton.

**Funding acquisition:** John O'Hara, Nick London.

**Investigation:** Thomas Hughes.

**Methodology:** Thomas Hughes, Alan Burton, Stacey Emmonds.

**Project administration:** John O'Hara, Stacey Emmonds.

**Resources:** John O'Hara.

**Supervision:** John O'Hara, Nick London, Stacey Emmonds.

**Writing – original draft:** Thomas Hughes.

**Writing – review & editing:** Thomas Hughes, John O'Hara, Nick London, Stacey Emmonds.

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
