## [Decision Letter · Decision Letter 0]

16 May 2023

PONE-D-23-08328Evaluating the implementation of injury prevention interventions in child and adolescent sports using the RE-AIM framework and CERT: A systematic reviewPLOS ONE

Dear Dr. Hughes,

Thank you for submitting your manuscript to PLOS ONE. After careful consideration, we feel that it has merit but does not fully meet PLOS ONE’s publication criteria as it currently stands. Therefore, we invite you to submit a revised version of the manuscript that addresses the points raised during the review process.

ACADEMIC EDITOR: Dear authors,

The authors already performed an interesting work and deserve great merit. Even so, it is still possible to improve the quality of the manuscript.

Please check all comments made by reviewers and address them point by point.

Thank you

We look forward to receiving your revised manuscript.

Kind regards,

Rafael Franco Soares Oliveira

Academic Editor

PLOS ONE

Additional Editor Comments:

Dear authors,

The authors already performed an interesting work and deserve great merit. Even so, it is still possible to improve the quality of the manuscript.

Please check all comments made by reviewers and address them point by point.

Thank you

Reviewers' comments:

Reviewer's Responses to Questions

**Comments to the Author**

1. Is the manuscript technically sound, and do the data support the conclusions?

Reviewer #1: Yes

Reviewer #2: Yes

Reviewer #3: Yes

2. Has the statistical analysis been performed appropriately and rigorously? 

Reviewer #1: N/A

Reviewer #2: Yes

Reviewer #3: Yes

3. Have the authors made all data underlying the findings in their manuscript fully available?

Reviewer #1: Yes

Reviewer #2: Yes

Reviewer #3: Yes

4. Is the manuscript presented in an intelligible fashion and written in standard English?

Reviewer #1: Yes

Reviewer #2: Yes

Reviewer #3: Yes

5. Review Comments to the Author

Reviewer #1: Thank you for the opportunity to review this manuscript. This is a very informative and well written manuscript looking at the quality of reporting across studies within youth sports injury prevention. I have some overarching points for consideration, which flow across the introduction and discussion. Additionally, I have included a few specific points.

Consider a rename on the title – as you are looking at all dimensions of RE-AIM, the ‘evaluation of implementation’ may not be a true representation of what your manuscript is capturing. Perhaps a simple removal of the word implementation and swapping in impact? Likewise, the reference to components of implementation is referenced throughout the manuscript. As you are also looking at reach, effectiveness, adoption and maintenance, it may be more appropriate to use an alternate word here to differentiate between the ‘implementation’ dimension of RE-AIM and program implementation overall. Readers from outside the field may be confused by the interchangeable nature of the terms.

This manuscript would also benefit from the inclusion of reference to internal and external validity as it applies to the included studies. For example, within you conclusion you note the increase reporting of effectiveness, which implies the studies are focused more on internal validity – whilst compromising on external validity (adoption, maintenance). This may have something to do with the phase in which many of these studies are, with early-stage study design (pilot, feasibility) often focusing more so on the effectiveness (unless a concerted effort to design for dissemination is made) rather than translation potential – when we start to consider adoption, implementation and maintenance aspects. This point on trial design is an important one, and the inclusion of internal and external validity related to trial type would strengthen this point.

Line 514: external validity introduced – this should be strengthened to highlight potential reasoning for findings

Line 538: Great consideration for future research and may tie in with my previous comments regarding trial phase.

Lines 554-557: This is fantastic. This is what I believe could be weaved in to my above point to help describe the reasoning for the % reporting across the RE-AIM dimensions. The sports injury programs included were likely in earlier phases along this continuum, so may not have considered the more ‘external validity’ focused metrics, and rather focused on effectiveness and some components of implementation.

Lines 682-691: This examples of trial phase/design and the data that is included is further evidence of my point above in regards to the focus of a study dependent on the phase in which it is in. The dissemination study of ‘Activate’ focuses more on elements of external validity (*i.e., adoption and maintenance as you have noted’ rather than effectiveness – as this was shown in a preceding study. This point is one that your manuscript would benefit from throughout – being able to highlight potential reasoning for the lack of adoption and maintenance reported due to the current focus of the literature in this review being focused more on efficacy/effectiveness than dissemination/translation of interventions.

Lines 708-711: Excellent sentence on the importance of policy for long term maintenance. This is an importance consideration I am glad you have included. This important consideration also ties into designing for dissemination and gaining buy in from delivery agents to update and modify injury prevention programs as they travel along the translation continuum. Your discussion on trial design considerations based of the work of Finch and Donaldson throughout lines 725-740 is excellent. This notion would benefit from being included in the introduction to preclude this point (i.e., mention of the variety of trial design and how these may impact on translation potential.

Reviewer #2: First of all, the reviewer would like to thank the authors for their work and efforts in trying to improve sports science knowledge. The article is evaluating the implementation of injury prevention interventions in child and adolescent

sports using the RE-AIM framework and CERT. Overall, the a systematic review is well designed and well-written, with a great introduction proposing the usefulness of the topic and a clear outline of the research question.

Reviewer #3: Review

Evaluating the implementation of injury prevention interventions in child and adolescent sports using the RE-AIM framework and CERT: A systematic review

I welcome studies that introduce novelty and applicability the important of exercise-based injury prevention interventions in child and adolescent sports (aged under 19 years) against the ‘Reach’, ‘Effectiveness’, ‘Adoption’, ‘Implementation’, ‘Maintenance’ (RE-AIM) framework and Consensus of Exercise Reporting Template (CERT), to ascertain level of reporting for the components of implementation. In fact, I am open to be persuaded to deep understand this relationship.

Hence, I am some sympathy with the author's intentions. In addition, the authors provide a decent description of the systematic review process, and some proposals for future research. The topic represents contemporary interest, and the scope of the work is appropriate for PlosO One. I think it is very important to conduct studies like this one, because in many occasions injury prevention programmes have demonstrated effectiveness in child and adolescent sports, provided compliance is adequate. However, little is known about the factors which relate to their implementation and whether the prevention programmes have been adopted and maintained in the long-term. Therefore, it is necessary to bring to light studies whose aim is to review what has been published previously on this topic in a rigorous way. This review is particularly important because, as the authors indicate, the publications on this topic are increasingly, so it is necessary to group all these papers to have a better understanding of the results obtained previously.

From my point of view, the main strength of the manuscript is the methodology. The authors have followed properly the PRISMA guidelines. In addition, the systematic review has been included in the PROSPERO register. It would be interesting to check this register, but the authors do not show the link with the purpose to save the anonymity. Moreover, they have consulted the most relevant databases and the diagram flow shows the selection process.

Apart from this general commentary about the manuscript, more details of some parts of the manuscript (strengths, weakness and questions) are found hereafter.

Abstract

1. In the abstract should be indicated at the end as well as in the conclusions section some directions about this retrospective and perspectives futures.

Introduction

2. It is well written and structured. It is a good starting point to place the reader. However, it would be interesting if the bibliography was updated. For example, it is necessary that the new papers were added. For example, in another sports.

- The most recent updates should be mentioned in the introduction considering? check:

3. Please improve the last paragraph and connect the background section with the relation with the aims of review at the end.

Methodology

4. As it has been mentioned before, this section is the strongest part of the manuscript. The PRISMA methodology is clear. The decision of including qualitative and quantitative literature is a good point of the study. However, the authors stated that ´The search finished July 2022. In that sense, the search must be updated until now (May 2023). It will add more quality and interest to the manuscript.

5. Please simplify the figure 2 and 3. Truly is necessary? Can you add in the text.

Discussion

- The conclusions needs to be rewritten. Please reference to the results of other studies, but the theoretical and practical implications of the research are vaguely mentioned. It necessary improve it.

6. PLOS authors have the option to publish the peer review history of their article (what does this mean?). If published, this will include your full peer review and any attached files.

Reviewer #1: No

Reviewer #2: No

Reviewer #3: No

---

## [Author Response · Author response to Decision Letter 0]

21 Jun 2023

Dear Dr Oliveira, 

Thank you for agreeing to review and edit our manuscript, I have reviewed the comments made by yourself and the reviewers and have made changes to the manuscript accordingly. Firstly, I’d like to thank the reviewers for reviewing our manuscript and providing useful comments for improvement. 

Reviewer’s comments:

Reviewer #1: Thank you for the opportunity to review this manuscript. This is a very informative and well written manuscript looking at the quality of reporting across studies within youth sports injury prevention. I have some overarching points for consideration, which flow across the introduction and discussion. Additionally, I have included a few specific points.

Consider a rename on the title – as you are looking at all dimensions of RE-AIM, the ‘evaluation of implementation’ may not be a true representation of what your manuscript is capturing. Perhaps a simple removal of the word implementation and swapping in impact? Likewise, the reference to components of implementation is referenced throughout the manuscript. As you are also looking at reach, effectiveness, adoption and maintenance, it may be more appropriate to use an alternate word here to differentiate between the ‘implementation’ dimension of RE-AIM and program implementation overall. Readers from outside the field may be confused by the interchangeable nature of the terms.

• In line with reviewer one’s comments regarding the title I have changed implementation to impact, as I agree that it describes the purpose of the review well and reduces the confusion with the ‘implementation’ domain of the RE-AIM framework.

This manuscript would also benefit from the inclusion of reference to internal and external validity as it applies to the included studies. For example, within you conclusion you note the increase reporting of effectiveness, which implies the studies are focused more on internal validity – whilst compromising on external validity (adoption, maintenance). This may have something to do with the phase in which many of these studies are, with early-stage study design (pilot, feasibility) often focusing more so on the effectiveness (unless a concerted effort to design for dissemination is made) rather than translation potential – when we start to consider adoption, implementation and maintenance aspects. This point on trial design is an important one, and the inclusion of internal and external validity related to trial type would strengthen this point.

• I agree with the need to include more reference to internal and external validity, therefore it has been addressed in the introduction and discussion. 

• I have included the reviewers comment regarding trial phase and emphasized that it may explain some of the distortion towards effectiveness measures as opposed to longer term measures.

Line 514: external validity introduced – this should be strengthened to highlight potential reasoning for findings

• Thank you for this point, I have strengthened the argument behind external validity and related it to the findings of the review

Line 538: Great consideration for future research and may tie in with my previous comments regarding trial phase.

• Thank you for this, I had not considered trial phase as importantly as I should. I have added comments into both the introduction and discussion.

Lines 554-557: This is fantastic. This is what I believe could be weaved in to my above point to help describe the reasoning for the % reporting across the RE-AIM dimensions. The sports injury programs included were likely in earlier phases along this continuum, so may not have considered the more ‘external validity’ focused metrics, and rather focused on effectiveness and some components of implementation.

• I have tried to weave external vs internal validity and trial phase into the discussuion.

Lines 682-691: This examples of trial phase/design and the data that is included is further evidence of my point above in regards to the focus of a study dependent on the phase in which it is in. The dissemination study of ‘Activate’ focuses more on elements of external validity (*i.e., adoption and maintenance as you have noted’ rather than effectiveness – as this was shown in a preceding study. This point is one that your manuscript would benefit from throughout – being able to highlight potential reasoning for the lack of adoption and maintenance reported due to the current focus of the literature in this review being focused more on efficacy/effectiveness than dissemination/translation of interventions.

• Thank you again for this point, I have tried to make the distinction between early phase and later phase research. I have also tried to emphasise that studies are not being penalised for their lack of long term data, as they may be publishing later.

Lines 708-711: Excellent sentence on the importance of policy for long term maintenance. This is an importance consideration I am glad you have included. This important consideration also ties into designing for dissemination and gaining buy in from delivery agents to update and modify injury prevention programs as they travel along the translation continuum. Your discussion on trial design considerations based of the work of Finch and Donaldson throughout lines 725-740 is excellent. This notion would benefit from being included in the introduction to preclude this point (i.e., mention of the variety of trial design and how these may impact on translation potential.

• Thank you for these points, I have brought the work by Donaldson and Finch into the introduction to enhance the background. 

Reviewer two comments:

Reviewer #2: First of all, the reviewer would like to thank the authors for their work and efforts in trying to improve sports science knowledge. The article is evaluating the implementation of injury prevention interventions in child and adolescent

sports using the RE-AIM framework and CERT. Overall, the a systematic review is well designed and well-written, with a great introduction proposing the usefulness of the topic and a clear outline of the research question.

• Thank you for these comments. 

Reviewer three comments:

Reviewer #3: Review

Evaluating the implementation of injury prevention interventions in child and adolescent sports using the RE-AIM framework and CERT: A systematic review

I welcome studies that introduce novelty and applicability the important of exercise-based injury prevention interventions in child and adolescent sports (aged under 19 years) against the ‘Reach’, ‘Effectiveness’, ‘Adoption’, ‘Implementation’, ‘Maintenance’ (RE-AIM) framework and Consensus of Exercise Reporting Template (CERT), to ascertain level of reporting for the components of implementation. In fact, I am open to be persuaded to deep understand this relationship.

Hence, I am some sympathy with the author's intentions. In addition, the authors provide a decent description of the systematic review process, and some proposals for future research. The topic represents contemporary interest, and the scope of the work is appropriate for PlosO One. I think it is very important to conduct studies like this one, because in many occasions injury prevention programmes have demonstrated effectiveness in child and adolescent sports, provided compliance is adequate. However, little is known about the factors which relate to their implementation and whether the prevention programmes have been adopted and maintained in the long-term. Therefore, it is necessary to bring to light studies whose aim is to review what has been published previously on this topic in a rigorous way. This review is particularly important because, as the authors indicate, the publications on this topic are increasingly, so it is necessary to group all these papers to have a better understanding of the results obtained previously.

From my point of view, the main strength of the manuscript is the methodology. The authors have followed properly the PRISMA guidelines. In addition, the systematic review has been included in the PROSPERO register. It would be interesting to check this register, but the authors do not show the link with the purpose to save the anonymity. Moreover, they have consulted the most relevant databases and the diagram flow shows the selection process.

Apart from this general commentary about the manuscript, more details of some parts of the manuscript (strengths, weakness and questions) are found hereafter.

• I appreciate this reviewer taking the time to review the manuscript in such detail. I have addressed all of the specific comments below. 

Unfortunately, the anonymity means we cannot provide the PROSPERO number at this time. 

Abstract

1. In the abstract should be indicated at the end as well as in the conclusions section some directions about this retrospective and perspectives futures.

• Thank you, I have reviewed the abstract and emphasized the recommended future directions for research at the end of the conclusion

Introduction

2. It is well written and structured. It is a good starting point to place the reader. However, it would be interesting if the bibliography was updated. For example, it is necessary that the new papers were added. For example, in another sports.

- The most recent updates should be mentioned in the introduction considering? check:

• Thank you, I have added additional studies on different sports in the introduction. I believe this is what you were meaning. 

3. Please improve the last paragraph and connect the background section with the relation with the aims of review at the end.

• I have tried to improve the last paragraph and provide a better link between the background and the aim. 

Methodology

4. As it has been mentioned before, this section is the strongest part of the manuscript. The PRISMA methodology is clear. The decision of including qualitative and quantitative literature is a good point of the study. However, the authors stated that ´The search finished July 2022. In that sense, the search must be updated until now (May 2023). It will add more quality and interest to the manuscript.

• The searches have been updated for May 2023, with amendments made to the statistics, results and figures. 

• Figure 1 (PRISMA flowchart) has been updated accordingly

5. Please simplify the figure 2 and 3. Truly is necessary? Can you add in the text.

• I have attempted to make figure 2 easier to read by using greyscale colouring on the different domains and have removed figure 3 as I agree with the reviewer that it is unnecessary.

Discussion

- The conclusions needs to be rewritten. Please reference to the results of other studies, but the theoretical and practical implications of the research are vaguely mentioned. It necessary improve it.

• Thank you, I have re-written the conclusion, making reference to previous reviews and strengthening the implications of the study. 

Many thanks for all comments,

Dr Thomas Hughes

---

## [Decision Letter · Decision Letter 1]

11 Jul 2023

Evaluating the impact of injury prevention interventions in child and adolescent sports using the RE-AIM framework and CERT: A systematic review

PONE-D-23-08328R1

Dear Dr. Thomas Hughes,

We’re pleased to inform you that your manuscript has been judged scientifically suitable for publication and will be formally accepted for publication once it meets all outstanding technical requirements.

Kind regards,

Rafael Franco Soares Oliveira

Academic Editor

PLOS ONE

Additional Editor Comments (optional):

Dear authors,

Congratulations! My recommendation is to accept!

Best regards

Reviewers' comments:

Reviewer's Responses to Questions

**Comments to the Author**

1. If the authors have adequately addressed your comments raised in a previous round of review and you feel that this manuscript is now acceptable for publication, you may indicate that here to bypass the “Comments to the Author” section, enter your conflict of interest statement in the “Confidential to Editor” section, and submit your "Accept" recommendation.

Reviewer #1: All comments have been addressed

2. Is the manuscript technically sound, and do the data support the conclusions?

Reviewer #1: Yes

3. Has the statistical analysis been performed appropriately and rigorously? 

Reviewer #1: N/A

4. Have the authors made all data underlying the findings in their manuscript fully available?

Reviewer #1: Yes

5. Is the manuscript presented in an intelligible fashion and written in standard English?

Reviewer #1: Yes

6. Review Comments to the Author

Reviewer #1: Well done on a thorough and comprehensive actioning of my previous review suggestions - this manuscript is reading very well with careful consideration of the potential for findings based on the literature.

Excellent work!

7. PLOS authors have the option to publish the peer review history of their article (what does this mean?). If published, this will include your full peer review and any attached files.

Reviewer #1: No

---

## [Editor Report · Acceptance letter]

14 Jul 2023

PONE-D-23-08328R1 

Evaluating the impact of injury prevention interventions in child and adolescent sports using the RE-AIM framework and CERT: A systematic review 

Dear Dr. Hughes:

I'm pleased to inform you that your manuscript has been deemed suitable for publication in PLOS ONE. Congratulations! Your manuscript is now with our production department. 

Kind regards, 

on behalf of

Prof Rafael Franco Soares Oliveira 

Academic Editor

PLOS ONE